# Patient-Reported Outcome Measures After Botulinum Toxin for Temporomandibular-Related Myalgia: A Prospective Study

**DOI:** 10.3390/jcm14217494

**Published:** 2025-10-23

**Authors:** Martijn van Soest, Lianne Remijn, Igor Tak, Egbert van der Hoeve, Laurens Koppendraaier, Maurits de Ruiter

**Affiliations:** 1Department of Oral and Maxillofacial Surgery, Diakonessenhuis, 3582 KE Utrecht, The Netherlandslkoppendraaier@diakhuis.nl (L.K.); 2Academy of Paramedical Studies, HAN University of Applied Sciences, 6525 EN Nijmegen, The Netherlands; lianne.remijn@han.nl; 3Fysiotherapie Utrecht Oost, 3581 WD Utrecht, The Netherlands; igor.tak@gmail.com

**Keywords:** botulinum toxin-A, temporomandibular disorders, myalgia, masticatory muscle pain, quality of life, mandibular function

## Abstract

**Introduction:** Botulinum toxin-A (BTX-A) injections are regularly used to treat temporomandibular disorders (TMD). However, consensus regarding the long-term efficacy of BTX-A for TMD-related myalgia remains lacking. This pragmatic, practice-based clinical study aimed to evaluate the Patient-Reported Outcome Measures of pain, health status, quality of life, and function after BTX-A injections in patients with TMD-related myalgia. **Methods:** This prospective cohort study included 35 patients with TMD-related myalgia who received BTX-A injections in the masseter and temporalis muscles. The Visual Analogue Scale for pain, the EQ-5D-3L for health status, the Oral Health Impact Profile-14 for oral health-related quality of life, the Mandibular Function Impairment Questionnaire for function and the maximum interincisal opening were assessed before treatment and at one, three and six months follow-up. **Results**: Patients reported a statistically significant and clinically relevant reduction in pain (*p* < 0.001), improvement of health status (*p* ≤ 0.003), and oral health-related quality of life (*p* < 0.001) at one-month follow-up, which remained present at three and six months post-treatment. Self-reported mandibular function and active and passive mouth opening showed no significant change over all time points. **Conclusions**: In this pragmatic cohort, BTX-A injections in the masseter and temporalis muscles seem to improve pain and oral health-related quality of life in patients with TMD-related myalgia within one month and show effects lasting up to six months, while mandibular function did not improve.

## 1. Introduction

Temporomandibular disorders (TMD) describe a group of disorders involving the temporomandibular joint (TMJ), masticatory muscles, and associated structures [1] and are reported to affect 10% to 30% of the global population [2]. Etiological factors contributing to TMD can be psychological, biological, and biomechanical [3]. According to the worldwide-accepted diagnostic criteria for temporomandibular disorders (DC/TMD), TMD are divided into two subgroups: articular disorders with signs and symptoms related to the TMJ, and pain related disorders with signs and symptoms related to muscular pain (myalgia) and headache [4].

In most cases of TMD, myalgia of the masticatory muscles is observed, which is frequently accompanied by restricted jaw opening and impaired oral function [5,6]. Consequently, TMD-related myalgia is associated with a reduced quality of life, psychological disorders (e.g., depression and anxiety), and may result in chronicity [7]. Current management for TMD-related myalgia includes the avoidance of triggers, behavioural techniques, decreasing jaw muscle activity through a soft diet, physical therapy, dental review for occlusal splints, and simple analgesia [1]. However, the long-term efficacy of these therapeutic modalities remains unclear, resulting in the search for other therapeutic interventions [8]. When TMD-related myalgia is persistent and does not respond to these conservative treatment options, therapy with Botulinum toxin-A (BTX-A) may be considered [1,6].

There is a growing use of BTX-A injections in the masticatory muscles as an intervention for TMD-related myalgia [9]. BTX-A results in muscle relaxation by temporarily blocking the release of acetylcholine from presynaptic cholinergic nerve terminals. The injected muscle remains paralyzed until new synaptic connections are formed through sprouting [10]. Additionally, BTX-A is reported to have antinociceptive effects by blocking the release of inflammatory mediators, such as substance P and glutamate [10,11]. The effects of BTX-A are reported to last two to four months [12]. Due to these muscle-relaxing and possible analgesic effects, BTX-A has gained increased interest as a possible treatment for TMD-related myalgia [13].

Studies have shown short-term improvements in Patient-Reported Outcome Measures (PROM’s) (up to three months) in pain intensity, function, and quality of life after BTX-A injections in the masseter and temporalis muscles in patients with TMD-related myalgia [14,15,16]. A systematic review by Thambar et al. [1] reported mixed short-term results: four studies showed significant pain reduction up to one or two months [17,18,19,20], while two others found no effect at three months [21,22].

Long-term outcomes (>three months) remain unclear [1,23]. One randomized controlled trial (RCT) reported lower pain scores at 12 months compared to other treatments [24]. A meta-analysis by Machado et al. [25] found no difference between BTX-A and placebo at three and six months suggesting that placebo responses may contribute to the observed effects, complicating the interpretation of BTX-A efficacy. An umbrella review by De la Torre Canales et al. [26] concluded that BTX-A is more effective than placebo for pain, but not superior to standard care, and highlighted potential adverse effects such as muscle atrophy and jawbone changes. Only one included study assessed outcomes beyond six months [27]. Importantly, most studies focused on pain, neglecting oral health-related quality of life [25,26]. A study by Li et al. [28] reported a reduction in pain intensity at six months following BTX-A treatment for TMD-related pain. However, Li et al. [28] included a relatively small sample size and a heterogeneous population with myogenous and arthrogenic TMD as well as bruxism-related complaints, which may limit the generalizability specifically to TMD-related myalgia. Moreover, while Li et al. [28] primarily evaluated pain reduction, the impact on quality of life and self-reported functional outcomes was not addressed.

The recent literature has emphasized that current evidence remains fragmented and that the clinical application of BTX-A for TMD-related myalgia should primarily be regarded as an adjuvant therapy or as a last treatment alternative [26,29]. Nevertheless, consensus regarding the efficacy of BTX-A for TMD-related myalgia remains lacking due to the inconsistent diagnostic criteria, variable dosages, potential side effects, and limited follow-up data [1,10,13,26,28]. Therefore, studies with extended follow-up that evaluate multidimensional outcomes such as pain, quality of life, and mandibular function are needed to guide clinical decision-making regarding BTX-A use in TMD-related myalgia.

Furthermore, by adopting a design that reflects practice-based clinical treatment conditions, where BTX-A is typically used as an adjunctive rather than a standalone therapy, this study aims to provide clinically relevant insights.

The aim of this study was to evaluate the clinical course of patients with TMD-related myalgia treated with BTX-A injections over a six-month period, focusing on pain intensity, health status, quality of life, and mandibular function within a pragmatic context. The hypothesis tested was that patients receiving BTX-A due to TMD-related myalgia would report a reduction in pain and increase in quality of life and mandibular function after one, three, and six months.

## 2. Materials and Methods

The study design was a prospective observational cohort study. The Medical Research Ethics Committees United provided a non-WMO (Medical Research Involving Human Subject Act) waiver (W23.212) prior to commencement of this study. All participants were informed and gave their written consent prior to participating, according to the requirements of the Declaration of Helsinki (World Medical Association, 2013).

### 2.1. Participants

Participants were recruited between January and August 2024 from the Department of Oral and Maxillofacial Surgery (OMS) at the Diakonessenhuis, Utrecht, The Netherlands.

Inclusion criteria for participation were the following:Adults with TMD-related myalgia in accordance with the DC/TMD;Acceptance of BTX-A injections as treatment;Ability to understand and complete questionnaires in Dutch.

Exclusion criteria were the following:Systemic inflammatory and connective tissue diseases (e.g., rheumatoid arthritis, ankylosing spondylitis, psoriatic arthritis);Surgery in the TMJ area < 12 months ago;Pain of dental origin;Limited cognitive functioning (not allowed to make medical decisions);Use of muscle relaxants or aminoglycoside antibiotics;A history of allergic reactions to BTX;Pregnancy and lactation.

All measurements were conducted between January 2024 and February 2025.

Prior BTX-A treatment was not an exclusion criterion, which may have influenced expectations and treatment response.

### 2.2. Study Procedure

Participants who were willing to enrol in the study received an information letter. Those who agreed to participate provided written informed consent prior to inclusion in the study. During the first visit, participants completed a standardized form detailing their demographic information, health history, previous treatments for TMD-related myalgia, and current symptoms. The researcher then checked the inclusion and exclusion criteria. Prior to the BTX-A injection, participants rated their current pain on a Visual Analogue Scale (VAS) [30]. Next, they completed the European Quality of Life 5 Dimensions 3 Level Version (EQ-5D-3L) [31], the Oral Health Impact Profile-14 (OHIP-14) [32], and the Mandibular Function Impairment Questionnaire (MFIQ) [33]. Then, the researcher measured the maximum active and passive mouth opening according to the DC/TMD.

### 2.3. Description of Measuring Instruments and Questionnaires

The VAS, which measures pain intensity, consists of a 100 mm line with two set points: 0 (indicating no pain) and 10 (indicating worst imaginable pain). The participants had to indicate their current level of pain by placing a mark on the line [30]. The minimal clinical difference in patients with chronic TMD-related pain is 19.5 mm [34].

The EQ-5D-3L questionnaire was used to assess the participants health status [31]. The participants rated their health on a 3-point Likert scale across five dimensions: mobility, self-care, usual activities, pain/discomfort, and anxiety/depression. The EQ-5D index, which ranges from less than 0 (worse health state) to 1 (perfect health), was calculated based on the Dutch population. Participants were also asked to rate their current general health status on a VAS-scale, ranging from 0, the worst possible health, to 100, the best possible health [31].

The OHIP-14 questionnaire assesses the oral health-related quality of life with answer options on a 5-point Likert scale, ranging from 0 (never) to 4 (very often). The total OHIP-14 score ranges from 0 to 56, with higher scores indicating more severe outcomes [32]. The smallest detectable difference (SDD) is 5 points [32,35].

Mandibular function was assessed by the sum score of the 17-item Dutch translation of the MFIQ [33]. Each item is scored on a 5-point Likert scale with score 0 indicating no difficulty and score 4 being extremely difficult to perform a mandibular task, resulting in a total score ranging from 0 to 68. Higher scores are indicative of increased levels of functional impairment. The SDD of the MFIQ is 10 points [33].

Maximal mouth opening was measured as the distance between the incisal edges of the central incisors in millimetres. This measurement was conducted using a metal ruler for both the active (AMO) and passive (PMO) mouth opening according to the DC/TMD [4]. The mean of three measurements was used [36]. The SDD in patients with painful restriction of the TMJ is 3 mm [36].

All measures were assessed prior to the BTX-A injection (T0) and at one (T1), three months (T2), and six months (T3) post-treatment by the same researcher (M.A.). The researcher was not involved in any treatment session. All measurements were conducted without the presence of the healthcare practitioner providing the injections to avoid observer bias due to the patient–doctor relationship. At all post-treatment measurements, the exclusion criteria were re-evaluated by asking the participants if they had other treatments for their TMD-related myalgia.

### 2.4. BTX-A Injections

All participants received a solution of BTX-A (Xeomin, Merz Pharma GmbH, Frankfurt am Main, Germany) in their masseter and temporalis muscles (total of 100 units reconstituted with 4cc unpreserved 0.9% sodium chloride). The injection was administered by the same maxillofacial surgeon (M.d.R.). The participants were asked to clench their teeth to determine the anatomical position of the masseter and temporal muscles with palpation. Intramuscular BTX-A injections were administered on both sides at the location of the masseter muscle (80 units), distributed in three different locations, and temporalis muscle (20 units), distributed in one location (Figure 1).

This technique was similar to other studies using BTX-A for the masseter and temporal muscles [17,20,37].

### 2.5. Statistical Analyses

All data were considered as metric values and were tested for normality of distribution using the Shapiro–Wilk Test and visual inspection of the histograms. Normally distributed data (*p* > 0.05) were presented as mean ± standard deviation, while non-normally distributed data were presented as median and (interquartile range 25–75%). Differences between T0, T1, T2, and T3 were analyzed using a repeated-measures ANOVA model for normally distributed data, and the Friedman test was applied in case of non-normally distributed data. In cases where significant outcomes were found, post hoc tests with Bonferroni corrections for multiple comparisons were applied.

A priori power analysis (G*Power 3.1) determined that a total of 30 participants were needed to achieve sufficient power (>90%) for detecting a pre–post difference on the VAS (Cohen’s d = 0.73) at α = 0.05. This power analysis assumes a known population mean of 65 mm with a standard deviation of 11 mm on the VAS [38]. The anticipated mean VAS for the study population was projected at 57 mm. This power calculation allows for an attrition rate of 10%. The alpha level for statistical significance was set at 0.05. All statistical analysis were performed using SPSS Statistics version 28.0.1.0 (IBM, Armonk, NY, USA).

## 3. Results

A total of 52 patients with TMD-related myalgia were invited to participate. Of these, 44 (85%) participated in this study. Nine participants were excluded during the study. Three patients were lost to follow-up, three patients received a new BTX-A treatment before the final follow-up, two participants became pregnant, and one participant was diagnosed with dental-related pain, which prevented them from further participation according to the exclusion criteria (Figure 2).

The 35 remaining participants (32 female, 3 male) completed all measurements. The characteristics of the participants are summarized in Table 1. All participants had previously received treatments for their TMD-related myalgia, including oral splint therapy, physical therapy, medication, or previous BTX-A injections.

Concurrent conservative therapies were recorded at baseline and monitored throughout follow-up. Minor individual changes were observed over time (Appendix B, Table A1), but these did not affect the primary outcome results.

Outcomes for the VAS, EQ-5D-3L, OHIP-14, MFIQ, AMO, and PMO at T0, T1, T2, and T3 are presented in Table 2.

The median VAS scores revealed a statistically significant difference (*χ*^2^ = 25.435, *p* < 0.001) over time. Post hoc analysis (Table 3a) revealed significant reductions in median pain at T1 (*Z* = −4.38, *p* < 0.001), T2 (*Z* = −3.58, *p* < 0.001), and T3 (*Z* = −4.13, *p* < 0.001), compared with T0. No significant differences in median pain were observed between T1, T2, and T3.

The median health status scores, as measured by the EQ-5D index scores, showed a statistically significant difference (*χ*^2^ = 14.598, *p* = 0.002) over time. Post hoc analysis (Table 3a) revealed a significant increase in health status at T1 (*Z* = −2.94, *p* = 0.003), T2 (*Z* = −3.43, *p* < 0.001), and T3 (*Z* = −3.01, *p* = 0.003), compared with T0. No significant differences in health status were observed between T1, T2, and T3. There were no significant mean differences in the EQ VAS over time (F (3, 102) = 1.751, *p* = 0.161).

The median OHIP-14 scores showed a significant difference over time (*χ*^2^ = 43.659, *p* = < 0.001). Post hoc analysis (Table 3a) revealed a significant median improvement of oral health-related quality of life at T1 (*Z* = −4.88, *p* < 0.001), T2 (*Z* = −4.47, *p* < 0.001), and T3 (*Z* = −4.27, *p* < 0.001), compared with T0. No significant differences in oral health-related quality of life were observed between T1, T2 and T3.

The median MFIQ scores showed a significant difference over time (*χ*^2^ = 8.672, *p* = 0.034). However, post hoc analysis with Bonferroni corrections (Table 3a) revealed no significant pairwise differences (*p* > 0.008). Similarly, the mean AMO showed a significant increase over time (F (2.156, 73.298) = 3.476, *p* = 0.033) but no significant pairwise differences during post hoc analysis (*p* > 0.008) (Table 3b). The mean PMO showed no significant increase over time (F (2.167, 73.507) = 2.334, *p* = 0.1).

One adverse effect has been reported during this study. A participant experienced mild asymmetry when smiling two days after receiving the BTX-A injections, which resolved completely within 12 weeks. No other adverse effects were reported.

## 4. Discussion

This prospective study evaluated the outcomes of BTX-A injections on pain, health status, oral health-related quality of life, and mandibular function in patients with TMD-related myalgia. We observed a significant and clinically relevant reduction in pain, as well as an improvement in health status and oral health-related quality of life at one month post-treatment, which was sustained until six months. No significant improvement was found in mandibular function or active and passive mouth opening. These findings support the positive outcomes of BTX-A injections for TMD-related myalgia with regard to pain and oral health-related quality of life up to six months, while mandibular function outcomes remain ambiguous.

These findings align with previous research demonstrating short-term improvements in pain and quality of life after BTX-A treatment [16,17,19,21,39,40] and extend these results to six months. A systematic review by Li et al. [28] showed similar results with a significant reduction in pain six months after BTX-A compared to placebo. However, the placebo-controlled, cross-over trial by Sitnikova et al. [40] showed that both placebo and BTX-A injections significantly reduced pain intensity and pain related disability at three and four months post-treatment. The discrepancy between the study of Li et al. [28] and Sitnikova et al. [40] may be explained by the lower dosage of BTX-A (50 units) but also reflects broader methodological issues such as heterogeneity in diagnostic criteria, variation in BTX-A dosage protocols, and the influence of placebo effects [29,40]. These findings highlight the need for standardized treatment protocols and further research comparing BTX-A with other therapeutic modalities and placebo.

No significant difference in mandibular function was observed across MFIQ, AMO, and PMO outcomes, consistent with previous studies [21,22]. While De la Torre Canales et al. [27] reported significant improvements in AMO and PMO after six months post BTX-A treatment, they found no significant improvements after one month. The delayed improvement in mouth opening suggests that these changes may be due to the normalization of mandibular function over time rather than directly being induced by the effects of BTX-A. Additionally, the assessment of the mandibular function may have been affected by the paralyzing effects of BTX-A, potentially resulting in higher impairment scores on the MFIQ due to reduced maximal biting forces when consuming tough and hard foods [41,42]. In light of the conflicting evidence on the efficacy of BTX-A in improving mandibular function, inclusion criteria for BTX-A injections in TMD-related myalgia should consider factors such as pain severity and quality of life.

All participants in this study had previously undergone conservative treatments such as physical therapy, oral splint use, or medication before receiving BTX-A, consistent with current recommendations [26]. However, there is no clear consensus on when conservative treatments should be considered insufficient, or when treatment with BTX-A should be initiated. While concurrent therapies were monitored, occlusal and prosthetic factors were not assessed. Although occlusal abnormalities have been associated with TMD, evidence for a causal relationship and specifically for their role in TMD-related myalgia remains unclear [43].

The treatment dosage of BTX-A varies widely in the literature [1,13,28]. Li et al. [28] showed that a bilateral dose of 60–100 units might be an optimal choice for treating pain in TMD-related myalgia. This variability underscores the need for standardized clinical guidelines and treatment steps defining dosage and injection protocols to ensure both efficacy and safety.

Repeated BTX-A injections are common due to their limited duration of effect (two to four months) [10,12,44], yet evidence on the efficacy and safety of repeated treatments of BTX-A for TMD-related myalgia is scarce. In this study, 37% of the participants had previously received BTX-A treatments, and three participants required repeated BTX-A due to an increase in pain levels. Future studies should evaluate the effects of repeated BTX-A treatments on pain, quality of life, and mandibular function, while also considering potential cumulative risks such as jawbone changes [26]. Long-term data on repeated injections are needed to determine whether treatment efficacy is maintained or diminishes over time. Cost-effectiveness should also be considered, particularly when repeated treatments offer limited additional benefit, and the cumulative costs may outweigh the clinical gains.

We acknowledge several limitations in our study. First, 37% of the participants had received prior BTX-A treatment. This may have introduced bias through altered expectations of efficacy or potential physiological carry-over effects, which could have influenced the outcomes and limited the generalizability. Moreover, TMD-related myalgia has a higher prevalence in women than in men [2]. Given that 91% of our study population was female, the findings may have limited external validity for the male population. Participants were not asked to discontinue concurrent conservative therapies such as splint use, physical therapy, or medication, reflecting a pragmatic design but introducing potential confounding. This approach mirrors routine clinical practice, where BTX-A is commonly used as an adjunctive rather than a standalone therapy, but it limits attribution of outcomes solely to BTX-A. Three patients were excluded as they required additional BTX-A treatment for severe pain, which may have led to the exclusion of high pain scores. Furthermore, the total scores of the OHIP-14 and MFIQ were treated as continuous variables in the analysis, consistent with previous research [16,45,46], although the individual items of these instruments are ordinal. Lastly, as this was a prospective observational cohort study, no causal conclusions can be drawn regarding the effects of BTX-A for TMD-related myalgia.

Despite these limitations, the study has several strengths. The pragmatic design reflects routine clinical practice, enhancing the clinical relevance of the results. Moreover, the six-month follow-up period contributes to understanding the long-term outcomes of BTX-A treatment for TMD-related myalgia. Furthermore, the sample size (*n* = 35) met the required threshold based on the power calculation, enhancing the reliability of the findings. Additionally, by including quality of life, the results broaden the scope of outcomes beyond pain and functional impairment in TMD-related myalgia. Given the limited knowledge on the clinical outcomes of quality of life after BTX-A, these findings may provide valuable insights into its clinical application in treating TMD-related myalgia.

Future research should prioritize high-quality randomized controlled trials with follow-up periods exceeding three months and a more balanced gender distribution to better assess BTX-A effectiveness. Comparative studies evaluating BTX-A against other treatment modalities, such as physical therapy, oral appliances, or pharmacological interventions, are needed to establish a more comprehensive treatment approach for TMD-related myalgia. Moreover, future studies with sufficient power should also consider stratifying outcomes based on prior BTX-A treatment and concurrent conservative therapies to enhance clinical transferability and guide the development of more personalized treatment approaches for patients with TMD-related myalgia. The lack of data on the cost-effectiveness of BTX-A for TMD-related myalgia highlights the need for further investigation in this area in clinical decision-making.

Given the current findings, BTX-A injections could be a valuable treatment option for reducing pain and improving health status and oral health-related quality of life in patients with TMD-related myalgia up to six months post-treatment. The lack of improvement in mandibular function should be taken into consideration when evaluating BTX-A treatment for TMD-related myalgia.

## Figures and Tables

**Figure 1 jcm-14-07494-f001:**
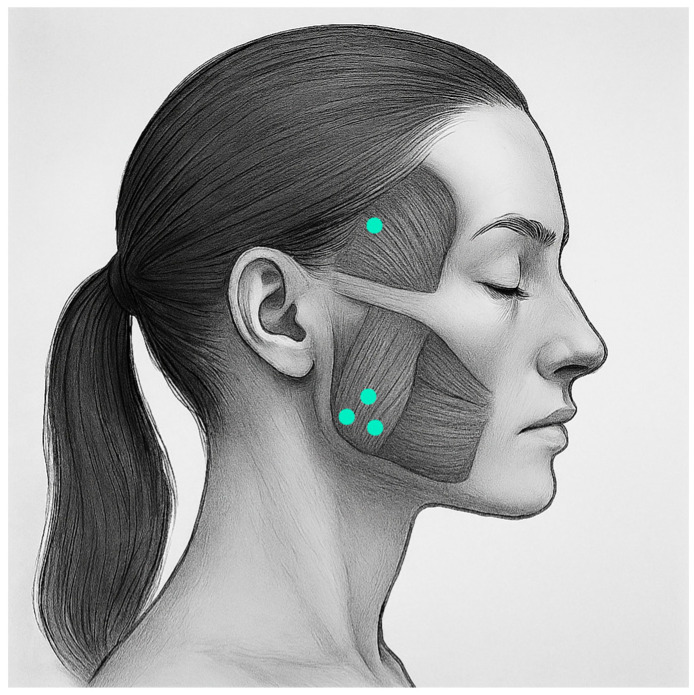
Location of BTX-A injections.

**Figure 2 jcm-14-07494-f002:**
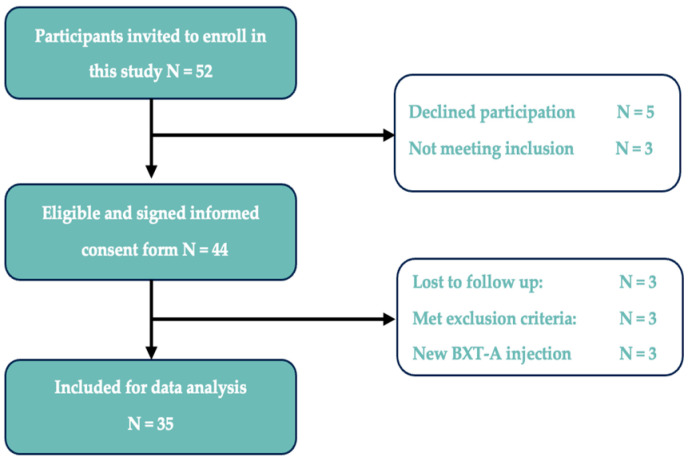
Flow chart of study inclusion/exclusion.

**Table 1 jcm-14-07494-t001:** Participants characteristics.

Female *n* (%)	32 (91)
Male *n* (%)	3 (9)
Age in years mean ± SD (range)	41 ± 15 (23–79)
Duration complaints in months median (IQR)	60 (24–120)
Previous BTX-A injection *n* (%)	13 (37)
Use of oral splint *n* (%)	28 (80)
Physical therapy *n* (%)	32 (91)
Prescribed medication *n* (%)	7 (20)

Abbreviations: *n*, number; SD, standard deviation; IQR, interquartile range 25–75%.

**Table 2 jcm-14-07494-t002:** Outcomes at baseline (T0), 1 month (T1), 3 months (T2), and 6 months (T3) (*n* = 35).

Variable	T0	T1	T2	T3
VAS (0–100)	58 (40–68)	27 (16–49)	31 (11–57)	28 (13–53)
EQ-5D Index	0.719 (0.252–0.811)	0.807 (0.683–1.000)	0.807 (0.687–0.843)	0.807 (0.687–1.000)
EQ VAS	69 ± 17	74 ± 15	70 ± 16	74 ± 14
OHIP-14	16 (11–22)	9 (5–13)	7 (4–12)	7 (2–12)
MFIQ	13 (5–27)	11 (7–18)	9 (4–20)	8 (2–14)
AMO	44 ± 10	45 ± 9	46 ± 9	46 ± 8
PMO	48 ± 9	49 ± 9	49 ± 9	50 ± 8

Data are presented as median (IQR 25–75%) or as mean ± SD. Abbreviations: SD, standard deviation; IQR, interquartile range; VAS, Visual Analogue Scale; EQ-5D, EuroQol 5 dimensions; OHIP-14, Oral Health-related Quality of Life evaluation; MFIQ, Mandibular Function Impairment Questionnaire; AMO, active mouth opening; PMO, passive mouth opening.

**Table 3 jcm-14-07494-t003:** (**a**) Pairwise comparisons of outcome measures between T0, T1, T2, and T3 ^a^. (**b**) Pairwise comparisons of outcome measures between T0, T1, T2, and T3.

(a)						
Variable						
		Median Difference	Z			*p*
VAS (0–100)	T0–T1	31	−4.83			<0.001 *
	T0–T2	27	−3.58			<0.001 *
	T0–T3	30	−4.13			<0.001 *
	T1–T2	4	−0.55			0.584
	T1–T3	1	−0.01			0.993
	T2–T3	3	−0.72			0.943
EQ-5D index	T0–T1	0.088	−2.97			0.003 *
	T0–T2	0.088	−3.43			<0.001 *
	T0–T3	0.088	−3.01			0.003 *
	T1–T2	0.000	−1.68			0.092
	T1–T3	0.000	−1.34			0.180
	T2–T3	0.000	−0.76			0.448
OHIP-14	T0–T1	7	−4.88			<0.001 *
	T0–T2	9	−4.47			<0.001 *
	T0–T3	9	−4.27			<0.001 *
	T1–T2	2	−1.11			0.266
	T1–T3	2	−1.12			0.261
	T2–T3	0	−0.40			0.687
MFIQ	T0–T1	2	−1.01			0.313
	T0–T2	4	−1.89			0.059
	T0–T3	5	−2.44			0.015
	T1–T2	2	−0.96			0.340
	T1–T3	3	−1.62			0.105
	T2–T3	1	0.81			0.418
**(b)**						
**Variable**					**95% CI for Difference**	
		**Mean Difference**	**Std. Error**	***p*** **^b^**	**Lower Bound**	**Upper Bound**
AMO	T0–T1	0.83	0.74	1.000	−2.91	1.25
	T0–T2	1.51	0.73	0.277	−3.56	0.54
	T0–T3	2.23	0.95	0.150	−4.89	0.43
	T1–T2	0.69	0.51	1.000	−2.10	0.73
	T1–T3	1.40	0.78	0.492	−3.59	0.79
	T2–T3	0.71	0.53	1.000	−0.76	2.19

(**a**) Abbreviations: ^a^, Wilcoxon signed ranks test; VAS, Visual Analogue Scale; EQ-5D, EuroQol 5 dimensions; OHIP-14, Oral Health-related Quality of Life evaluation; MFIQ, Mandibular Function Impairment Questionnaire; * *p* < 0.008 (Bonferroni correction). (**b**) Abbreviations: CI, confidence interval; ^b^, Bonferroni correction for multiple comparisons; AMO, active mouth opening.

## Data Availability

The data that support the findings of this study are available from the corresponding author upon reasonable request.

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
