# Peer review of "Patient-Reported Outcome Measures After Botulinum Toxin for Temporomandibular-Related Myalgia: A Prospective Study"

_jcm, 2025, doi:10.3390/jcm14217494_

Round 1

Reviewer 1 Report

Comments and Suggestions for Authors

General Evaluation

The manuscript entitled “Patient-Reported Outcome Measures After Botulinum Toxin for Temporomandibular-Related Myalgia: A Prospective Study” presents a prospective cohort investigation assessing the effects of botulinum toxin-A (BTX-A) on pain, health status, oral health-related quality of life, and mandibular function in patients diagnosed with TMD-related myalgia.

The study is well-conceived, methodologically robust, and addresses a significant clinical issue with promising implications for everyday practice. The manuscript is clearly articulated and systematically organized, and the findings are pertinent to the ongoing discourse regarding the role of BTX-A in the management of TMD-related myalgia.

Nonetheless, certain revisions are necessary.

Major Comments

Conceptualization and Rationale

The introduction aptly summarizes the epidemiology, clinical burden, and prevailing controversies concerning the application of BTX-A. However, it requires more explicit positioning of the present investigation within the framework of high-level evidence and current debates.

Specifically, recent systematic reviews and controlled studies should be incorporated into the discussion to enhance the rationale and balance the interpretative perspective. The following publications should be cited and elaborated upon:

Dental and Medical Problems: 10.17219/dmp/200127

These contributions address methodological considerations, safety issues, and the hierarchy of evidence concerning BTX-A in myogenous TMD and ought to be integrated into both the Introduction and the Discussion sections.

Review of Literature

The literature review is reasonably comprehensive but somewhat selective. To ensure balance, the limitations identified in recent meta-analyses and umbrella reviews—such as heterogeneity of diagnostic criteria, limited follow-up duration, placebo effects, and safety concerns—should be more critically emphasized.

The brief mention of the clinical controversy regarding cost-effectiveness and repeated injections warrants expansion, given the study’s relevance to long-term outcomes.

Methods

The methodological approach is appropriate, featuring clear inclusion and exclusion criteria, standardized outcome measures, and a sufficiently justified sample size based on power analysis.

However, the fact that 37% of patients had prior BTX-A treatment introduces potential bias, which should be explicitly acknowledged as a limitation in both the Methods and Discussion sections.

Results

The results are presented systematically with appropriate statistical analyses. The authors are to be commended for including both pain and quality of life measures, which are often neglected in this domain.

Figures and tables are suitable; however, some could be optimized through more concise labeling (for instance, combining OHIP-14 and EQ-5D outcomes into a single composite figure).

Discussion

The discussion maintains balance but could benefit from a more critical examination of discordant findings within the literature.

The conclusion should adopt a more cautious tone, emphasizing that although BTX-A improved pain and quality of life for up to six months, mandibular function outcomes remain ambiguous. Additionally, high-quality randomized controlled trials with extended follow-up are necessary.

Inclusion of the suggested references will facilitate contextualization of the findings within the broader scientific corpus.

Clinical Relevance

The manuscript possesses strong clinical applicability, given its focus on PROMs that directly reflect patient-centered outcomes.

Nonetheless, the discussion should underscore practical considerations for clinicians, such as patient selection criteria, expected benefit duration, the necessity for repeat injections, and potential adverse effects.

Minor Comments

The abstract could be refined to more precisely report the unchanged functional outcomes (MFIQ, AMO, PMO), as these are clinically significant.

Minor typographical errors should be addressed (e.g., “botolinum” should be corrected to "botulinum” in the keywords).

Uniformity in citation style should be maintained, as some references (e.g., references 36, 37, 39) exhibit inconsistent formatting of author initials.

Recommendation: Major Revision

The manuscript is promising and offers valuable prospective data concerning PROMs following BTX-A injections in TMD-related myalgia. Upon integration of recent high-quality references, a more balanced discussion of conflicting evidence, and minor editorial corrections.

Author Response

Author's Reply to the Review Report (Reviewer 1)

General response
Thank you very much for taking the time to review this manuscript. We appreciate your constructive feedback and suggestions, which have helped us to improve the quality and clarity of the paper. Please find below our point-by-point responses. All revisions and corrections are highlighted in the re-submitted manuscript (track changes).

Comment 1 (Conceptualization and Rationale)

The introduction aptly summarizes the epidemiology, clinical burden, and prevailing controversies concerning the application of BTX-A. However, it requires more explicit positioning of the present investigation within the framework of high-level evidence and current debates.

Specifically, recent systematic reviews and controlled studies should be incorporated into the discussion to enhance the rationale and balance the interpretative perspective. The following publications should be cited and elaborated upon:

Dental and Medical Problems: 10.17219/dmp/200127

These contributions address methodological considerations, safety issues, and the hierarchy of evidence concerning BTX-A in myogenous TMD and ought to be integrated into both the Introduction and the Discussion sections.

Response 1 (Conceptualization and Rationale)

Thank you for this valuable suggestion. We have revised the introduction and discussion to strengthen the rationale by incorporating recent reviews and studies including (2025; Dent Med Probl. doi:10.17219/dmp/200127). This addition in the introduction and discussion provides a more explicit positioning of our study within the context of current evidence and debates.

  • Change made: Introduction, page 2, paragraph 6, lines 92-97.
  • Change made: Discussion, page 8, paragraph 2, lines 294-299.

Comment 2 (Review of Literature)

The literature review is reasonably comprehensive but somewhat selective. To ensure balance, the limitations identified in recent meta-analyses and umbrella reviews—such as heterogeneity of diagnostic criteria, limited follow-up duration, placebo effects, and safety concerns—should be more critically emphasized.

Response 2 (Review of Literature)

We agree with this comment and have revised the literature review to include a more critical discussion of heterogeneity in diagnostic criteria, placebo effects, limited follow-up duration, and safety concerns.

  • Change made: Introduction, page 2, paragraph 5, lines 85-91.
  • Change made: Introduction, page 2, paragraph 6, lines 92–97.
  • Change made: Discussion, page 8, paragraph 2, lines 294-299.
  • Change made: Discussion, page 9, paragraph 4, lines 378-385.

Comment 3 (Review of Literature)

The brief mention of the clinical controversy regarding cost-effectiveness and repeated injections warrants expansion, given the study’s relevance to long-term outcomes.

Response 3 (Review of Literature)

We appreciate this insightful comment. We agree that the cost-effectiveness and repeated injections of BTX-A in TMD-related myalgia is an important aspect. However, as our study did not include an economic evaluation or an investigation of repeated injections, we considered an extensive discussion of this topic to be beyond the scope of the current manuscript. To address the reviewer’s point, the discussion acknowledges the lack of data on repeated injections and cost-effectiveness and highlighting the need for future research in this area.

  • Change made: Discussion, page 9, paragraph 4, lines 378-385.
  • Change made: Discussion, page 10, paragraph 1, lines 485-494

Comment 4 (Materials and Methods)

The methodological approach is appropriate, featuring clear inclusion and exclusion criteria, standardized outcome measures, and a sufficiently justified sample size based on power analysis. However, the fact that 37% of patients had prior BTX-A treatment introduces potential bias, which should be explicitly acknowledged as a limitation in both the Methods and Discussion sections.

Response 4 (Materials and Methods)

We agree with this comment, and this limitation is now explicitly acknowledged in both the Methods and Discussion sections.

  • Change made: Methods, page 3, paragraph 3, lines 135-136
  • Change made: Discussion, page 9, paragraph 4, lines 374-380

Comment 5 (Results)

The results are presented systematically with appropriate statistical analyses. The authors are to be commended for including both pain and quality of life measures, which are often neglected in this domain.

Figures and tables are suitable; however, some could be optimized through more concise labeling (for instance, combining OHIP-14 and EQ-5D outcomes into a single composite figure).

Response 5 (Results)

We thank the reviewer for this suggestion. We carefully considered combining the OHIP-14 and EQ-5D results into a single figure. However, because these instruments use different measurement scales and units, merging them would decrease the clarity and readability of the results. For this reason, we decided to present them separately to maintain transparency and interpretability.

  • Change made: Figure 3 (now Figure A.1.) was transferred to the Appendix to avoid redundancy with Table 2.

Comment 6 (Discussion)

The discussion maintains balance but could benefit from a more critical examination of discordant findings within the literature.

Response 6 (Discussion)

We thank the reviewer for this valuable comment. We have revised the discussion to provide a more critical analysis of the discordant findings in the literature. In particular, we now emphasize how methodological heterogeneity (diagnostic criteria, dosing protocols, study design) and placebo effects may explain conflicting outcomes across studies. We have also added references to strengthen this argument.

  • Change made: Discussion, page 8, paragraph 2, lines 294-299.

Comment 7 (Discussion)

The conclusion should adopt a more cautious tone, emphasizing that although BTX-A improved pain and quality of life for up to six months, mandibular function outcomes remain ambiguous. Additionally, high-quality randomized controlled trials with extended follow-up are necessary.

Response 7 (Discussion)

We agree and have revised the conclusion accordingly, emphasizing the ambiguity of functional outcomes, and the need for high-quality RCTs with extended follow-up.

  • Change made: Discussion, page 8, paragraph 1, lines 286-287.
  • Change made: Discussion, page 10, paragraph 5, lines 396-398.

Comment 8 (Discussion)

Inclusion of the suggested references will facilitate contextualization of the findings within the broader scientific corpus.

Response 8 (Discussion)

We agree and have added the suggested reference for contextualization and a broader scope.

  • Change made: References, page 14, references 28, lines 604-505.

Comment 9 (Clinical Relevance)

The manuscript possesses strong clinical applicability, given its focus on PROMs that directly reflect patient-centered outcomes.

Nonetheless, the discussion should underscore practical considerations for clinicians, such as patient selection criteria, expected benefit duration, the necessity for repeat injections, and potential adverse effects.

Response 9 (Clinical Relevance)

We agree and have expanded the discussion with practical considerations, explicitly addressing duration of benefit, repeated injections, and possible adverse effects.

  • Change made: Discussion, page 8, paragraph 2, lines 294-299.
  • Change made: Discussion, page 9, paragraph 5, lines 374-385.

Comment 10 (Minor Comments)

The abstract could be refined to more precisely report the unchanged functional outcomes (MFIQ, AMO, PMO), as these are clinically significant.

Response 10 (Minor Comments)

We thank the reviewer for this helpful comment. We have revised the abstract to explicitly state that mandibular function (MFIQ) and active and passive mouth opening (AMO, PMO) showed no significant changes over all time points. This makes the unchanged functional outcomes more precise and clinically clear.

  • Change made: Abstract, page 1, paragraph 1, line 26

Comment 11 (Minor Comments)

Minor typographical errors should be addressed (e.g., “botolinum” should be corrected to "botulinum” in the keywords).

Response 11 (Minor Comments)

We thank the reviewer for pointing this out. The typographical error has been corrected: “botolinum” has been changed to “botulinum” in the keywords.

  • Change made: Keywords, page 1, line 30

Comment 12 (Minor Comments)

Uniformity in citation style should be maintained, as some references (e.g., references 36, 37, 39) exhibit inconsistent formatting of author initials.

Response 12 (Minor Comments)

We thank the reviewer for this observation. The reference list has been carefully revised to ensure uniformity in citation style, and the inconsistencies in author initials (including references 36, 37, and 39) have been corrected.

  • Change made: References, page 12-14, lines 540-650

Comment 13 (Major revision)

The manuscript is promising and offers valuable prospective data concerning PROMs following BTX-A injections in TMD-related myalgia. Upon integration of recent high-quality references, a more balanced discussion of conflicting evidence, and minor editorial corrections.

Response 13 (Major Comments)

We sincerely thank the reviewer for the positive assessment and constructive feedback. We have carefully integrated high-quality references, expanded the introduction and discussion to provide a more balanced view of conflicting evidence, and made suggested editorial corrections. We believe these changes have strengthened the manuscript.

Reviewer 2 Report

Comments and Suggestions for Authors

This study is a simple, prospective study that evaluated patient-reported outcomes related to TMD-related myalgia treated with BTX-A injections, using validated patient surveys. Although the results are interesting, the study lacks originality, as its greatest strength is the 6-month assessment, which has already been reported in other studies, such as Li 2024. The objective should be rewritten to determine a more precise originality for the study. 
However, the study presents other problems and limitations related to its design, specifically the inclusion criteria and the way the results are expressed:
- An essential inclusion criteria should be the suspension of any other treatment for TMD-related myalgia, as the results currently obtained could not be attributed exclusively to BTX-A injections.
- Since this study is not a placebo-controlled trial, and since the results are not compared with those obtained with other types of treatment for myalgia, this study has limited clinical validity.  - The patients' dentition, dental arch, and occlusion were not taken into consideration when segmenting the sample: complete arch? Dental wear? Reduction in vertical dimension? Dental prosthesis? Acrylic prosthesis? Porcelain prosthesis? Implant-supported prosthesis? This aspect is extremely important due to the close relationship between occlusal disorders and temporomandibular disorders
- The results shown in Table 2 duplicate those shown in Figure 3. Choose only one way to present them
- Figure 3 should show statistically significant differences with asterisks and express the value of each p in the figure legend, thus excluding the  tables where statistical comparisons are expressed.

Author Response

Author's Reply to the Review Report (Reviewer 2)

General response
Thank you very much for taking the time to review this manuscript. We appreciate your constructive feedback and suggestions, which have helped us to improve the quality and clarity of the paper. Please find below our point-by-point responses. All revisions and corrections are highlighted in the re-submitted manuscript (track changes).

Comment 1 (Conceptualization and rationale)

This study is a simple, prospective study that evaluated patient-reported outcomes related to TMD-related myalgia treated with BTX-A injections, using validated patient surveys. Although the results are interesting, the study lacks originality, as its greatest strength is the 6-month assessment, which has already been reported in other studies, such as Li 2024. The objective should be rewritten to determine a more precise originality for the study. 

Response 1 (Conceptualization and rationale)

We thank the reviewer for this valuable comment and agree that the objective of the study required clarification to better reflect its originality and clinical relevance. In the revised manuscript, we have expanded the Introduction to clearly differentiate our study from previous research, particularly Li et al. (2024), and to emphasize its unique aspects. Specifically, we highlight that:

The current study focused exclusively on patients with TMD-related myalgia, whereas Li et al. included a heterogeneous population (myogenous TMD, arthrogenic TMD, and bruxism), which limits comparability.

The present study assessed a broader range of validated patient-reported outcomes, including quality of life, which were not extensively explored in previous work.

The study was conducted within a pragmatic and clinical setting, reflecting how BTX-A is commonly used as an adjunctive therapy rather than as a standalone treatment.

These additions are described in the abstract and introduction and the study aim has been reformulated to explicitly reflect these aspects.

  • Change made: Abstract, page 1, paragraph 1, lines 14-15.
  • Change made: Introduction, page 2, paragraph 4-5, lines 85-97.
  • Change made: Introduction, page 3, paragraph 1-2, lines 98-106.

Comment 2 (Design and methods)

However, the study presents other problems and limitations related to its design, specifically the inclusion criteria and the way the results are expressed:
- An essential inclusion criteria should be the suspension of any other treatment for TMD-related myalgia, as the results currently obtained could not be attributed exclusively to BTX-A injections.

Response 2 (Design and methods)

We thank the reviewer for this comment. We fully agree that concurrent therapies may influence treatment outcomes. However, this study was designed pragmatically to reflect routine clinical practice, in which BTX-A is typically administered as an adjunctive rather than a standalone therapy. Therefore, ongoing conservative treatments such as physical therapy, splint use, or medication were not discontinued. This approach aimed to capture outcomes in a real-world clinical setting and to enhance the clinical relevance of the findings. Changes in ongoing therapies were monitored and are now described in the results section. We have also clarified this point in the Discussion and acknowledged it as a limitation of the study.

  • Change made: results, page 6, paragraph 2, lines 231-233.
  • Change made: Discussion, page 9-10, paragraph 6, lines 387-392.
  • Change made: Appendix B, page 12, Table B.1.

Comment 3 (Design and methods)

Since this study is not a placebo-controlled trial, and since the results are not compared with those obtained with other types of treatment for myalgia, this study has limited clinical validity.

Response 3 (Design and methods)

We thank the reviewer for this valuable insight. We agree that, as this study was not placebo-controlled and did not include a comparison with other treatment modalities, causal inferences regarding the efficacy of BTX-A cannot be made. However, this study was designed to reflect the clinical practice, in which BTX-A is commonly used as an adjunctive therapy rather than a standalone intervention. This approach allows the findings to provide clinically relevant insight into treatment outcomes under routine care conditions. We have clarified this point in the Discussion section and explicitly acknowledged this as a limitation.

  • Change made: Discussion, page 9, paragraph 5, lines 382-385
  • Change made: Discussion, page 9, paragraph 6, lines 387-392
  • Change made: Discussion, page 9, paragraph 6, lines 396-398

Comment 4 (Review of literature)

The patients' dentition, dental arch, and occlusion were not taken into consideration when segmenting the sample: complete arch? Dental wear? Reduction in vertical dimension? Dental prosthesis? Acrylic prosthesis? Porcelain prosthesis? Implant-supported prosthesis? This aspect is extremely important due to the close relationship between occlusal disorders and temporomandibular disorders

Response 4 (Review of literature)

We thank the reviewer for this insightful comment. We acknowledge that dentition, occlusion, and dental arch conditions could play a role in the pathophysiology of temporomandibular disorders (TMD). However, recent reviews indicate that while occlusal factors such as malocclusion, tooth loss, and bruxism are associated with TMD in general, the evidence for a causal or predominant role, particularly in TMD-related myalgia, remains limited and inconsistent [Lekaviciute & Kriauciunas, 2024 DOI:10.7759/cureus.54130; Lassmann et al., 2025 DOI:10.1111/jerd.13303; Pascu et al., 2025 DOI: 10.3390/medicina61050791]. We therefore consider our decision not to stratify by occlusal variables justified. We decided to describe this choice in the discussion.

  • Change made: Discussion, page 9, paragraph 3, lines 365-368

Comment 5 (Results)

The results shown in Table 2 duplicate those shown in Figure 3. Choose only one way to present them.

Response 5 (Results)

We thank the reviewer for this helpful suggestion. We agree that Table 2 and Figure 3 presented overlapping information. To avoid redundancy and improve readability, Table 2 has been retained in the main text as it provides detailed statistical data, while Figure 3 (now Figure A1) has been moved to the Appendix as supplementary material.

  • Change made: Results, page 6, paragraph 3, lines 235.

Comment 6 (Results)

Figure 3 should show statistically significant differences with asterisks and express the value of each p in the figure legend, thus excluding the tables where statistical comparisons are expressed.

Response 6 (Results)

We thank the reviewer for this suggestion. Since the detailed statistical comparisons are already presented in Table 2, we decided to retain the table in the main text and move Figure 3 (now Figure A.1.) to the Appendix as supplementary material. The figure legend has been revised to indicate which variables showed significant differences, along with the corresponding p-values, to ensure clarity and transparency.

  • Change made: Appendix A, page 11, lines 522-526.

We sincerely thank the reviewer for their thorough evaluation and constructive feedback. The comments have significantly contributed to improving the clarity, structure, and scientific quality of the manuscript. We have carefully addressed each point and revised the text accordingly. We believe that the revised version provides a clearer rationale, better methodological transparency, and improved presentation of the results, enhancing both the scientific and clinical relevance of the study.

Round 2

Reviewer 1 Report

Comments and Suggestions for Authors

General Evaluation

The manuscript has been substantially revised. The authors have clarified the rationale, expanded the literature review, integrated recent high-level evidence, and acknowledged important limitations. The text is better balanced, and the conclusions are more cautious. Overall, the revision improved the scientific robustness and clinical value of the study. Some minor issues remain.

Assessment of Revisions

Conceptualization and Rationale

  • The introduction now situates the study within the context of recent systematic reviews and umbrella reviews.

  • The suggested reference (Dental and Medical Problems: 10.17219/dmp/200127 – Val et al., 2025) has been included and discussed, with emphasis on hierarchy of evidence and safety.

  • Remaining issue: although the rationale is stronger, the contrast between positive observational findings and limitations from meta-analyses could be emphasized more clearly in the introduction. A short paragraph synthesizing this controversy would help sharpen the focus.

Review of Literature

  • Expanded to include heterogeneity of diagnostic criteria, placebo response, and safety issues (e.g., muscle atrophy, bone remodeling).

  • Cost-effectiveness and repeated injections are now mentioned in the Discussion. However, this section could still be slightly expanded to highlight practical challenges of repeated cycles and uncertain cost-benefit ratio.

Methods

  • Inclusion/exclusion criteria remain robust.

  • Prior BTX-A exposure (37%) is now explicitly acknowledged in Methods and discussed as a limitation. Good improvement.

Results

  • Results are systematically presented, statistical analysis is sound.

  • Figures and tables are clear, but the OHIP-14 and EQ-5D remain presented separately. A composite figure would improve readability but is not essential.

  • Functional outcomes (MFIQ, AMO, PMO) are clearly shown as unchanged—this addresses the earlier request.

Discussion

  • More balanced: now cites discordant studies (Li et al. 2024 vs. Sitnikova et al. 2024).

  • Authors acknowledge heterogeneity, placebo effects, and lack of functional improvement.

  • Remaining issue: the negative finding on mandibular function deserves greater emphasis. Currently, it is mentioned but softened. This should be framed as a clinically important limitation.

  • The conclusion now calls for high-quality RCTs with long-term follow-up—this aligns with reviewer recommendations.

Clinical Relevance

  • Patient-centered outcomes are highlighted.

  • Practical aspects (patient selection, repeated injections, adverse events) are included, though the expected duration of benefit (3–6 months) should be explicitly stated in the clinical implications section.

Minor Comments

  • Abstract now specifies unchanged function outcomes.

  • Typographical error “botolinum” in keywords has been corrected.

  • Reference formatting is improved, though minor inconsistencies remain (author initials not uniform in refs 36–41). A final editorial check is needed.

English Grammar and Style

  • Overall improved, though some redundancies and awkward constructions remain (e.g., “This improvementthat sustained until six months”).

  • A careful language polish for conciseness and flow is recommended.

Final Recommendation: Minor Revision

The authors have satisfactorily addressed the major comments from the first review. The manuscript is now methodologically sound, clinically relevant.

Required minor revisions:

  1. Strengthen emphasis on lack of functional improvement (MFIQ, AMO, PMO) in the Discussion and Conclusion.

  2. Expand slightly on cost-effectiveness and repeated injection considerations in the Discussion.

  3. Ensure uniform reference formatting (author initials, journal abbreviations).

  4. Minor English editing for conciseness and removal of small typographical errors.

 Overall evaluation: The manuscript is much improved and close to acceptance. With these final refinements, it will make a solid contribution to the literature on BTX-A and TMD-related myalgia.

Author Response

Author's Reply to the Review Report round 2 (Reviewer 1)

General response
We sincerely thank the reviewer for the constructive and thoughtful feedback. We are pleased that the manuscript is considered much improved and close to acceptance. Below, we address each of the requested minor revisions point by point. All revisions and corrections are highlighted in the re-submitted manuscript (track changes).

Comment 1 (Conclusion and discussion)

Strengthen emphasis on lack of functional improvement (MFIQ, AMO, PMO) in the Discussion and Conclusion.

Response 1 (Conclusion and discussion)

Thank you for this valuable suggestion. We have strengthened the emphasis on the lack of improvement in mandibular function across MFIQ, AMO, and PMO measures in both the Discussion and Conclusion. The revised text highlights that while pain and quality of life improved, functional outcomes did not, underscoring the limited impact of BTX-A on mandibular function.

  • Change made: Abstract, page 1, conclusion, lines 29-30.
  • Change made: Discussion, page 8, paragraph 3, lines 305-306.
  • Change made: Discussion, page 10, paragraph 3, lines 403-305.

Comment 2 (Discussion)

Expand slightly on cost-effectiveness and repeated injection considerations in the Discussion.

Response 2 (Discussion)

We appreciate this insightful comment. The Discussion section has been expanded to include additional considerations regarding repeated BTX-A injections and their cost-effectiveness. We now note that long-term data are needed to determine whether efficacy diminishes over time and that the cumulative costs of repeated treatments may outweigh their clinical benefits.

  • Change made: Discussion, page 9, paragraph 4, lines 350-354.
  • Change made: Discussion, page 10, paragraph 2, lines 392-398.

Comment 3 (Reference)

Ensure uniform reference formatting (author initials, journal abbreviations).

Response 3 (Reference)

Thank you for noticing this detail. We thoroughly reviewed and standardized all references according to the journal’s guidelines, ensuring consistent author initials, journal abbreviations, and citation numbering throughout the manuscript.

  • Change made: References, page 13-14, lines 457-620.

Comment 4 (English)

Minor English editing for conciseness and removal of small typographical errors.

Response 4 (English)

Thank you for this recommendation. We performed a comprehensive language review to improve conciseness, grammar, and readability. Minor typographical and stylistic corrections were made throughout the text such as:

  • Change made: Introduction, page 3, paragraph 2, line 106.
  • Change made: Methods, page 5, paragraph 2, lines 183.
  • Change made: Methods, page 5, paragraph 2, lines 204.
  • Change made: Results, page 8, paragraph 4, lines 281.
  • Change made: Discussion, page 8, paragraph 1, lines 288.

We would like to thank the reviewer once again for the helpful and constructive feedback. We believe these final refinements have further improved the clarity, precision, and overall quality of the manuscript, strengthening its contribution to the literature on BTX-A and TMD-related myalgia.

Reviewer 2 Report

Comments and Suggestions for Authors

I greatly appreciate the authors' willingness to make all the changes and clarifications suggested in this manuscript. They have truly contributed enormously to improving the interpretation and clarity of the manuscript. While the originality of the results and conclusions as expressed here considerably improves the quality of the work. However, if the authors could stratify the results according to the primary type of treatment for TMD myalgia and according to whether or not prior treatment with BTX-A injections was used, the quality and clinical transferability of the results would be even greater.

Author Response

Author's Reply to the Review Report Round 2 (Reviewer 2)

General response
We sincerely thank the reviewer for the feedback. We greatly appreciate the recognition of our efforts to improve the clarity, interpretation, and originality of the manuscript. The thoughtful suggestion provided has been carefully considered and addressed as follows.

Comment 

I greatly appreciate the authors' willingness to make all the changes and clarifications suggested in this manuscript. They have truly contributed enormously to improving the interpretation and clarity of the manuscript. While the originality of the results and conclusions as expressed here considerably improves the quality of the work. However, if the authors could stratify the results according to the primary type of treatment for TMD myalgia and according to whether or not prior treatment with BTX-A injections was used, the quality and clinical transferability of the results would be even greater.

Response

We thank the reviewer for this insightful suggestion and fully agree that stratifying the results according to prior BTX-A treatment and type of primary conservative therapy would provide valuable additional insights.

However, given the limited sample size (N=35), such subgroup analyses would currently lack sufficient statistical power and likely induce misinterpretation of the results. Your comment has led us to add a statement in the discussion, noting that future studies with sufficient power should perform stratified analyses to improve clinical transferability and develop more personalized treatment strategies.

  • Change made: Methods, page 10, paragraph 2, lines 395-398.

We once again thank the reviewer for the constructive feedback and supportive comments.

We believe this additional clarification further strengthens the transparency and clinical relevance of the manuscript and appreciate the opportunity to refine it accordingly.
